# A Consolidated Saccharification, Fermentation, and Transesterification Process (cSFT) Converting Castor Oil to Biodiesel with Cellulose-Derived Ethanol

**DOI:** 10.3390/ijms262411902

**Published:** 2025-12-10

**Authors:** Ester Korkus Hamal, Gilad Alfassi, Dmitry M. Rein, Yachin Cohen

**Affiliations:** 1Department of Chemical Engineering, Technion-Israel Institute of Technology, Haifa 3200003, Israel; 2Department of Biotechnology Engineering, Braude College of Engineering, Karmiel 2161002, Israel; agilad@braude.ac.il

**Keywords:** cellulose, cellulase, bioprocess, emulsion, yeast, lipase, ^1^H NMR

## Abstract

Environmental and economic concerns due to the increasing use of fossil-based chemicals, especially fuel, may be alleviated by production of renewable fuels based on plant biomass, in particular, waste. Multistep cascades of enzymatic reactions are being increasingly sought to enhance the effectiveness of sustainable, environment-friendly processes. The biochemical transformation of lignocellulosic biomass and oils into fatty acid esters (“biodiesel”) involves biomass pretreatment, followed by polysaccharide hydrolysis and sugar fermentation to alcohol, either sequentially or simultaneously. Subsequent trans-esterification with waste or non-food-based oils is usually carried out in an organic solvent. Biocatalysis in aqueous emulsion offers significant advantages. This study presents a novel “one-pot” emulsion-based process for transforming unmodified cellulose and castor oil into biodiesel via hybridized yeasts with cellulose-coated micro-particles incorporating cellulolytic enzymes and lipases. The resultant consolidated bioprocess of saccharification, fermentation, and transesterification (cSFT) promotes effective substrate channeling and can potentially serve as a model for emulsion-based “one-pot” transformations of cellulose into valuable chemicals.

## 1. Introduction

The increasing consumption of fossil fuels has raised environmental and economic concerns, which have driven a call for urgent adoption of renewable alternative raw materials for fuels. A viable substitute is the biopolymer cellulose, the main component of biomass, as it is one of the most abundant and renewable materials on Earth [1]. Cellulose can be obtained from wood, cotton, and other plant fibers, or fungi, algae, and bacteria, as well as from plant-based waste [2,3,4,5]. Ethanol is then mainly produced via enzymatic hydrolysis of the cellulosic materials and fermentation of the extracted glucose [6,7,8]. Over the years, there has been substantial advancement in the conversion of lignocellulosic biomass to alcohols, owing to extensive research and technological developments devoted to improving and optimizing the biotechnological processes. These directions include immobilization of yeast cells [9,10,11,12,13,14], simultaneous saccharification and fermentation (SSF) [15,16,17], and metabolic engineering systems for bacteria [18,19,20], fungi [21] and yeasts [5,22,23] for consolidated bioprocessing, such as hydrolysis of biomass and conversion to alcohols [24] and oleochemicals [25,26]. Moreover, significant research has been conducted on conversion of ethanol into biodiesel by improving the efficiency of enzyme-catalyzed processes [21,22,23,24,25,26,27,28,29,30,31,32].

Multistep cascades of enzymatic reactions are being increasingly utilized as sustainable, environment-friendly synthetic pathways, which offer advantages such as mild conditions, efficiency, and selectivity [33,34,35,36]. In numerous technologically relevant applications, biphasic systems are employed to optimize the local conditions for enzymes operating on distinct substrates, particularly in situations where both aqueous and organic media are required [37]. These offer the advantages of high interfacial area and thermodynamic stability [38,39,40,41].

In previous studies, we reported the emulsifying capacity of unmodified cellulose [42]. The formed microcapsules had a unique inner structure comprising a hydrophobic core enveloped by an inner shell composed of aqueous cellulose hydrogel encapsulated by a more compact cellulose outer coating [43,44]. Cellulolytic enzymes adhere to the outer coating for effective cellulose hydrolysis [45]. Emulsion-based simultaneous saccharification and fermentation (eSSF) was achieved by addition of (engineered) yeasts to such an emulsion [46]. Lipases introduced to the aqueous emulsion medium spontaneously assembled at the inner core–hydrogel interface and catalyzed transesterification of fatty acids and alcohols in the hydrophobic core [47]. More specifically, the incorporated lipase effectively catalyzed the transesterification of castor oil in the core with ethanol dissolved in the aqueous medium, rather than simple hydrolysis, despite the abundance of water [48]. Moreover, we recently reported the self-assembled hybridization of yeasts (*S. cerevisiae*) with such cellulose-coated emulsion micro-particles incorporated with cellulolytic enzymes. The resulting system enabled simultaneous enzymatic hydrolysis of the cellulose in the micro-particle coating and yeast fermentation of the produced glucose to ethanol, with the proximity of the emulsion particles and yeasts ensuring efficient substrate channeling [49]. This report introduces and validates a novel one-pot process for transformation of cellulose into biodiesel by yeasts hybridized with cellulose-coated emulsion micro-particles incorporating cellulolytic enzymes and lipases. The system was designed to allow for consolidated saccharification, fermentation, and transesterification (cSFT) with effective substrate channeling. This cascade includes the enzymatic hydrolysis of the cellulose coating, production of ethanol by yeast fermentation of the released glucose, and generation of biodiesel at the inner micro-particle interface by lipase-catalyzed transesterification. A schematic representation of the reactions is depicted in Figure 1. This unique system holds the potential to serve as a micro-bioreactor for “one-pot” processes transforming cellulose into biodiesel as well as other valuable chemicals [50].

## 2. Results

An integrated system of emulsified castor oil micro-particles encapsulated with cellulose, incorporated with enzymes (cellulase, β-glucosidase, and lipase), hybridized with yeast, was generated by the method described previously [49]. The structure of the hybrid system was examined using a cryogenic scanning electron microscope (cryo-SEM). Figure 2 presents images recorded following partial sublimation of the vitrified water matrix, showing circular-shaped micro-particles and yeast cells. The background appearance in Figure 2a,b exhibits smooth and grainy textured regions, the former being bare glassy water and the latter due to fibrils of dispersed cellulose hydrogel particles. The micro-particles and yeast cells can be seen to be mostly embedded within the cellulose hydrogel. A sectioned yeast particle with adjacent cellulose-coated emulsion particles is depicted in Figure 2b. Imaging the same region using an energy-selective backscattered electron (EsB) detector (Figure 2c) provided contrast based on elemental composition. With such contrast, oil in the micro-particle core was distinguished as darker circular objects within brighter areas of cellulose hydrogel and water. The particles’ diameter evaluated by analysis of SEM images was broadly distributed between 0.2 and 2.4 µm (Appendix A). The number distribution of particle sizes, evaluated by light scattering measurements performed on the integrated system of micro-particles and yeasts at the reaction’s initiation, is presented as an overlay in Figure 2a. The number distribution presentation reflects the dimensions of the emulsion micro-particles, which are much more numerous compared with the yeasts, as discussed in the Appendix A. This size distribution correlated well with the cryo-SEM images (Figure 2a overlay).

The unique system was designed to function as a micro-bioreactor for a “one-pot” cSFT process transforming cellulose into biodiesel, as shown schematically in Figure 1. The cascade begins with transformation of glucose, produced by the hydrolysis of the cellulose of both the micro-particle coating and the dispersed hydrogel, into the yeast for fermentation to ethanol, and, subsequently, back to the particle for lipase-catalyzed trans-esterification of triacyl-glyceride (TAG) to fatty acid ethyl ester (FAEE). Close proximity of yeast and micro-particles is shown in the cryo-SEM image in Figure 3, taken after 48 h of reaction.

The structural and chemical changes occurring during FAEE production were followed over 96 h by cryo-SEM imaging, light scattering (LS) measurements, and ^1^H nuclear magnetic resonance (NMR) analysis. Figure 4 illustrates the progress of the cSFT process in the combined system. It can be observed that the free cellulose hydrogel was digested within 24 h, and the number of micro-particles was gradually reduced, indicating consumption of cellulose by enzymatic saccharification. After 72 h, the emulsion micro-particles were no longer detectable, having been completely consumed (Figure 4c). The LS measurements shown in Figure 4d are in qualitative accord with the cryo-SEM imaging. As discussed in the Appendix A, the number distributions after reaction durations of 24 h and 48 h were due to the abundance of micro-particles, whereas after 72 h, only the yeasts remained.

^1^H NMR analysis confirmed the generation of FAEE within the core of the cellulose-coated emulsion micro-bioreactors by the consolidated cSFT process, directly from cellulose. It is well known that lipases can catalyze the transformation of triglyceride oil in an ethanol/water solution either by hydrolysis to free fatty acids (FFAs) and glycerol or transesterification to FAEE, depending on the water content [51]. The progress of the cSFT process can be evaluated by following the transitions in the ^1^H NMR spectra of oil, FAEE, and FFAs obtained by the lipase-catalyzed reactions of castor oil in an aqueous emulsion, under conditions of water abundance, as presented in Figure 5. The spectrum of the reaction system containing the integrated cellulose-coated emulsion particles with yeast, at the initiation of the cSFT process before addition of cellulolytic enzymes, is shown in Figure 5a.

The regions between the dashed lines in Figure 5 indicate the characteristic peaks of castor oil. Conversion of the oil to FAEE was quantified using the method described by Sumit [52], calculated from integration of the relevant peaks by Appendix A. ^1^H NMR spectra of castor oil, FAEE, and FFAs, their chemical formulae, and the labeling of the protons relevant to the peaks used in the conversion calculation are also provided in the Appendix A. It can be observed in Figure 5a–e that the castor oil peaks in the range of 4.28–4.32 ppm (region A) diminished dramatically in the first 24 h, whereas the quartet FAEE peak in the range of 4.10–4.17 ppm emerged progressively with time. In this range, there was some overlap with remnants of the peaks attributed to the oil, which was taken into account in the equation of Sumit [52] (Appendix A). As the reaction progressed, the initial castor oil peaks in the range of 2.24–2.38 ppm (region B) transformed into overlapped peaks of FFAs and FAEE. The conversion of cellulose to FAEE validates our previous work, which demonstrated that lipase dispersed in the aqueous medium of a cellulose-coated emulsion can penetrate the coating and react with fatty esters [48]. Figure 5f presents the FAEE yield achieved after increasing durations of the cSFT process. As expected, the conversion of FAEE, as determined based on the amount of castor oil in the micro-particle core, increased with time and peaked at 34% after 96 h of the cSFT process. From this value, the transformation yield of cellulose in the micro-particles and in free hydrogel to FAEE was estimated at approximately 50%.

Figure 6 demonstrates the progress of the cSFT process by presenting the concentrations in the aqueous emulsion medium of the intermediate reaction products, glucose due to enzymatic cellulose hydrolysis, and ethanol by yeasts’ fermentation, and of the FAEE concentration in the organic phase. The glucose conversion rate was high in the initial 24 h and stabilized after 48 h. In parallel, the ethanol conversion rate was elevated during the first 48 h and then gradually decreased. The concentration of FAEE produced by lipase-catalyzed transesterification increased significantly during the initial 24 h, while during the rest of the reaction, it increased moderately. These reactions occurred simultaneously; there was generation and consumption of glucose and ethanol throughout the cSFT process. Moreover, the close proximity of yeasts and micro-particles [49], which served for efficient mass transfer, minimized the loss of reaction intermediates to the aqueous medium.

## 3. Discussion

Integration of yeast cells with cellulose-coated micro-particles presents a novel approach for bioreactors: one-pot conversion of cellulose directly to FAEE, which is conducted in an aqueous emulsion rather than in an organic solvent. This study provides proof-of-concept for the cSFT process in cellulose-coated emulsion micro-particles. This lies in sharp contrast with most previous studies, in which lipase-catalyzed transesterification took place in non-aqueous environments, to avoid competing hydrolysis [53]. Therefore, a unique aspect of this cSFT process is its stability and effectiveness in an aqueous environment. The lipases introduced into the aqueous emulsion medium spontaneously assemble at the inner particle interface between the TAG core and the inner hydrogel shell. This unique micro-environment provided effective catalysis of transesterification between ethanol and TAG in the particle core, rather than TAG hydrolysis, despite the abundance of water in this system. This novel cSFT scheme can, thus, benefit from the known advantages of a water-based process. These results highlight the significance of minimizing the dilution of intermediate compounds (glucose and ethanol) into the abundant aqueous medium. This is achieved by the novel hybrid configuration that facilitates close contact at micron-scale dimensions between all the system components, enabling optimal substrate channeling. Of note, the cSFT process was highly cost-effective, requiring a substantially lower amount of cellulase (3 FPU g^−1^ cellulose) compared with other works [54,55,56]. Such low enzyme loading also preserved the integrity of the micro-particle structure, as the cellulose coating was hydrolyzed concurrently with the fermentation and transesterification reactions. The critical balancing of the catalytic performance with the preservation of micro-particle morphology preserved the encapsulation of the castor oil core throughout the consolidated process, as well as its proximity to the yeasts that produced the intermediate ethanol. Indeed, loss of glucose and ethanol intermediates to the emulsion medium is the key factor affecting FAEE yield. Under optimal conditions, separation of the organic phase at the end of the process is expected to be facile due to complete cellulose hydrolysis. This study reports the proof-of-concept of the consolidated process and should be subject to further optimization.

To our knowledge, this is the first report of an innovative configuration of yeast integrated with emulsion micro-particles (coated by cellulose) loaded with enzymes that function as a bioreactor to convert cellulose to FAEE by effective substrate channeling [49]. While the substrate in the reported experiments was microcrystalline cellulose, the system can be applied to other forms of pretreated biomass. The system presented in this work demonstrates not only effective conversion of cellulose into biodiesel but also holds potential as a platform for synthesizing other valuable chemicals, enabling integration of biochemical reactions. It may be particularly relevant for engineered microorganisms with enhanced capabilities for biomass valorization [32,57].

## 4. Materials and Methods

### 4.1. Materials

The source of cellulose used in this study was micro-crystalline cellulose (MCC) powder obtained from Sigma Aldrich (Rehovot, Israel; polymerization degree: ~295, and particle size: ~20–160 µm, as provided by the supplier). The other chemicals, including 1-butanol, sodium hydroxide, deuterated chloroform (CDCl_3_), yeast (*Saccharomyces cerevisiae*), type II baker’s yeast, and β-glucosidase from *Aspergillus niger* were also obtained from Sigma Aldrich (Rehovot, Israel). Castor oil was purchased from Chen Samuel Chemicals (Haifa, Israel). The cellulytic enzyme Celluclast^®^ 1.5 L was obtained from Novozymes A/S (Bagsvaerd, Denmark), and Lipozyme^®^ TL was received from Tzamal-Medical (Petach Tikva, Israel).

### 4.2. Methods

#### 4.2.1. Preparation of Emulsion Integrated with *S. cerevisiae*

First, a suspension of cellulose hydrogel particles was prepared, as described previously [49], with minor changes. Molecularly dissolved cellulose solutions (4 wt. %) were obtained by mixing MCC in aqueous NaOH (7 wt. %) at room temperature and then in a cooling bath (−20 °C) using a mechanical stirrer at 500 rpm, until no crystalline cellulose was observed visually, as reported previously [49]. Coagulation of the cellulose solution was induced by the addition of deionized water without stirring. The coagulated hydrogel was rinsed multiple times until electrical conductivity measurements indicated removal of alkali traces (below 1 mS cm^−1^). A sodium acetate buffer solution at a pH of 4.8 (50 mmol L^−1^) was added to the hydrogel dispersion.

Castor oil was emulsified by the suspension of cellulose hydrogel particles to form cellulose-coated micro-particles. As described previously [49], castor oil served as a model oil due to its benefits, as it is non-toxic, miscible in alcohol, renewable, biodegradable, and, in particular, a non-edible oil [58,59]. The total cellulose content was 2 wt. %. Micro-particles were obtained by two steps: A pre-emulsion was prepared by mixing a cellulose hydrogel dispersion (~3.5 wt. % cellulose), castor oil at a cellulose–oil wt. ratio of 1:6, and water, using an IKA^®^ T-18 Ultra-Turrax^®^ mechanical homogenizer (IKA Works Inc., Wilmington, NC, USA) at 20,000 rpm for 5 min. The coarse emulsion was subjected to high-pressure homogenization (HPH) using a model LM-20 microfluidizer (Microfluidics, Westwood, MA, USA) at a homogenization pressure of 20,000 psi for 4 min. During HPH, the temperature was kept at around 40 °C by using ice. Yeasts (*S. cerevisiae*) were added to 10 mL emulsion samples to achieve a 1 wt. % concentration.

#### 4.2.2. Imaging by Cryogenic Scanning Electron Microscopy (Cryo-SEM)

An Ultra Plus FEG-SEM (Carl Zeiss, Jena, Germany) high-resolution cryogenic scanning electron microscope (cryo-HRSEM) was used to image the morphology of the micro-particles emulsion. It was equipped with a Schottky field-emission gun and a Bal-Tec VCT100 (Balzers, Liechtenstein) cold stage. The methods of cryogenic specimen preparation and their imaging were developed in the Technion Center for Electron Microscopy of Soft Matter and employed in a previous study [49]. About ~3 µL of the emulsion sample between two metallic carriers was subsequently vitrified by high-pressure freezing (HPF) (Leica EM ICE). Then, the frozen sample was transferred to a Leica EM ACE900 freeze fracture unit (Leica Microsystems, Wetzlar, Germany) via a pumped cryo-transfer shuttle, maintained at −170 °C. A rapidly cooled knife was used to fracture the frozen sample. The temperature was raised to −100 °C for 3.5 min to remove some of the ice by sublimation to expose structural features and improve contrast [60]. The fractured sample was transferred to the HRSEM for imaging. The samples were imaged at a low electron acceleration voltage (0.8–1 kV) and working distance (3–4.6 mm) to minimize charging. An Everhart–Thornley detector (SE2) and an in-column (InLens) detector for high-resolution surface information were used. In addition, elemental contrast was observed using an energy-selective backscattered (EsB) detector. The images were examined with imageJ 1.53, scientific image analysis software (U.S. National Institutes of Health, Bethesda, MD, USA).

#### 4.2.3. One-Pot Reaction and Bio-Conversion of Cellulose to Biodiesel

The cSFT process was performed in 15 mL Falcon tubes containing 10 mL emulsion samples of integrated yeast–micro-particle hybrids. Celluclast^®^ (3 FPU g^−1^ cellulose), β-glucosidase (3 CBU g^−1^ cellulose), and TL (120.3 KLU g^−1^) were added to these samples. The reaction was carried out in a shaker incubator at 40 °C and 200 rpm for 24, 48, 72, and 96 h. Samples for glucose analysis by the dinitrosalicylic acid (DNS) method, GC analysis, and ^1^H NMR analysis were collected.

#### 4.2.4. Nuclear Magnetic Resonance (NMR) Analysis

^1^H NMR spectroscopy was used to quantify the production of fatty acid ethyl ester (biodiesel) by the lipase-catalyzed transesterification reaction, as described previously [49]. The samples were centrifuged for 5 min at 6000 rpm, and 20 µL from the top phase was collected and dissolved in 680 µL of CDCl_3_ (with the chemical shift value at 7.26 ppm) in 5 mm NMR tubes at room temperature. ^1^H NMR (400 MHz) spectra were obtained with a Bruker-ARX 400 instrument (Bruker, Billerica, MA, USA).

#### 4.2.5. Dinitrosalicylic Acid (DNS) Method

The glucose concentration in the reacted cSFT samples was determined using the DNS method [61]. The DNS method is a colorimetric method based on the simultaneous oxidation and reduction of one mole of the aldehyde functional group and one mole of 3,5-dinitrosalycilic acid, respectively. The absorbance is considered a correlate of the glucose concentration in the samples. Quantification was performed using a calibration curve of known standards (575 nm).

#### 4.2.6. Gas Chromatography Analysis (GC)

Gas chromatography was used to quantify the ethanol concentrations achieved by the cSFT reactions, as described previously [49], except that in this case, butanol was used as the internal standard [62]. The samples were centrifuged for 5 min at 6000 rpm, and 5 mL of the clear solution and 100 µL of butanol were transferred into a calibration bottle. Finally, 800 µL of the sample was transferred into a glass vial and injected into a CP-3800 gas chromatograph (Varian Inc., Walnut Creek, CA, USA) equipped with a CP-8907 column and a flame ionization detector. A flow rate of 1.1 mL min^−1^ was used.

## 5. Conclusions

This study examined a novel “one-pot” process driven by cellulose-coated micro-particles incorporating cellulolytic enzymes, lipases, and yeast to directly transform cellulose and castor oil into biodiesel. The unique microenvironment ensured effective substrate channeling, despite the abundance of water in the system. ^1^H NMR analysis provided a quantitative determination of the consolidated bioprocess of saccharification, fermentation, and transesterification (cSFT), and essentially provided the first proof for an integrated water-based system that is stable and active for effective and direct conversion of cellulose to FAEE via a cSFT bioprocess. The presented system can potentially serve as a model for emulsion-based “one-pot” transformation of cellulose into valuable chemicals.

## Figures and Tables

**Figure 1 ijms-26-11902-f001:**
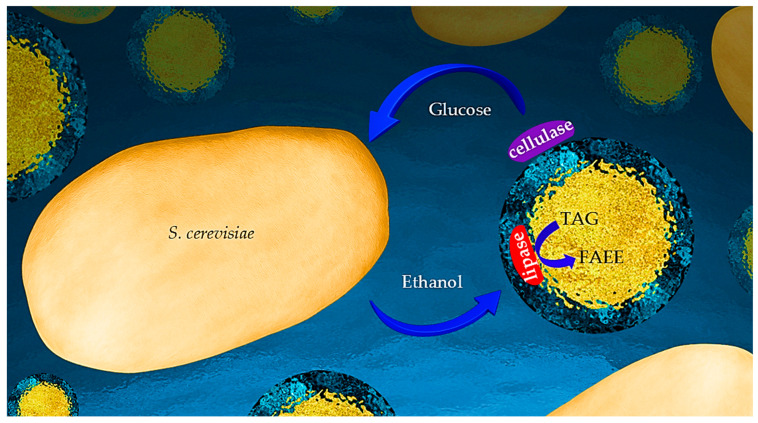
Scheme of the cSFT bioprocess. Cellulose-coated micro-particles, with a triacyl glyceride (TAG) core, incorporated with cellulolytic and lipase enzymes, are in close contact with yeast. The setup enabling a cascade of biochemical reactions is, thus, achieved: enzymatic cellulose hydrolysis and yeast fermentation of glucose and lipase-catalyzed TAG transesterification into fatty acid ethanol ester (FAFE).

**Figure 2 ijms-26-11902-f002:**
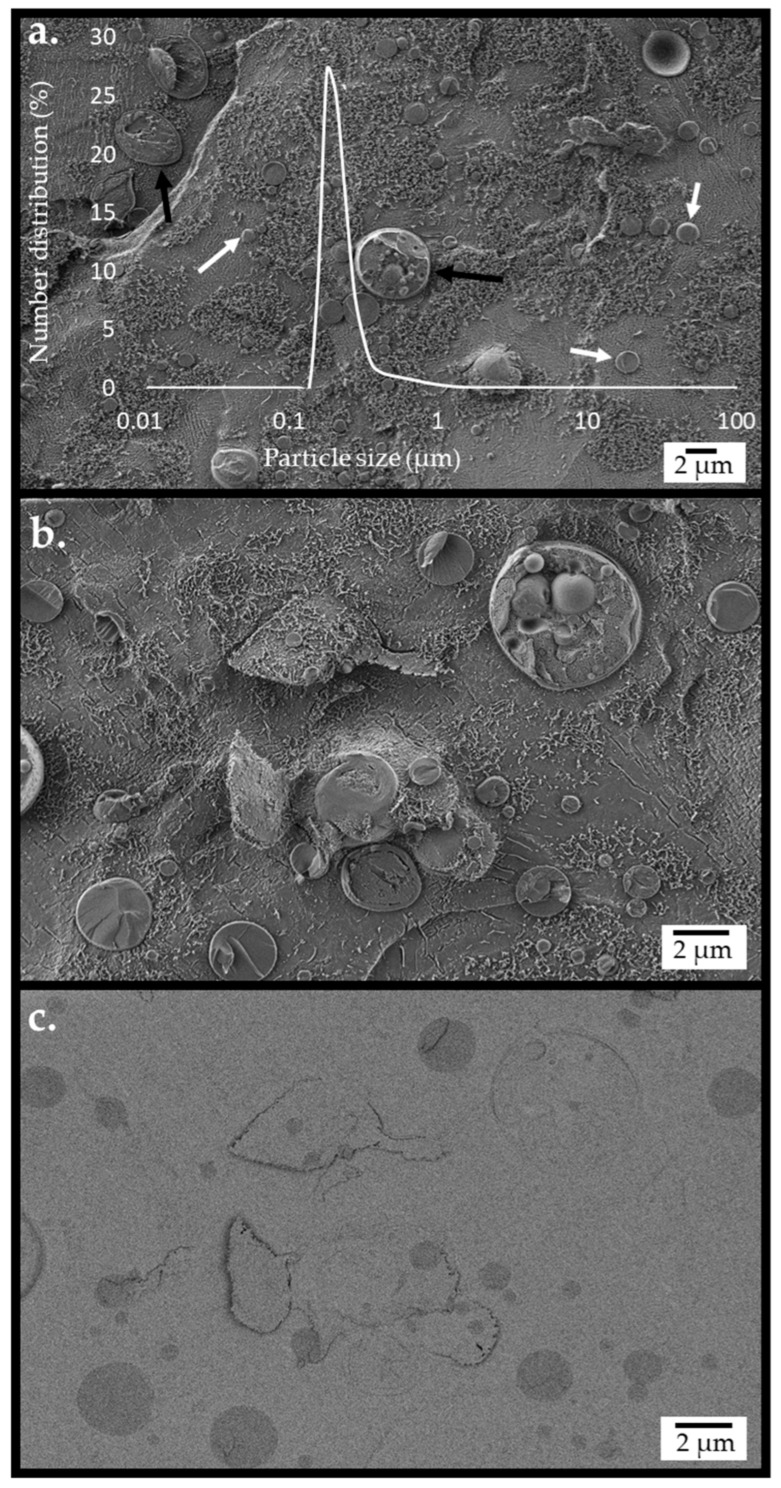
The structure of the emulsified cellulose-coated micro-particle integrated with dispersed *S. cerevisiae*, obtained by cryo-SEM imaging of the cryo-fractured surface of the vitrified emulsion. The emulsion fabrication conditions were as follows: a cellulose–castor oil wt. ratio of 1:6 and high-pressure homogenization (HPH) at 20,000 psi. The images were recorded using (**a**) secondary electron (SE2) and InLens detectors, overlaid with the particle size distribution by the number evaluated by light scattering at time 0. Arrows indicate a yeast cell (black) and micro-particles (white). (**b**) Images recorded using SE2 and InLens detectors at a higher resolution. (**c**) Images recorded using a high-resolution EsB detector.

**Figure 3 ijms-26-11902-f003:**
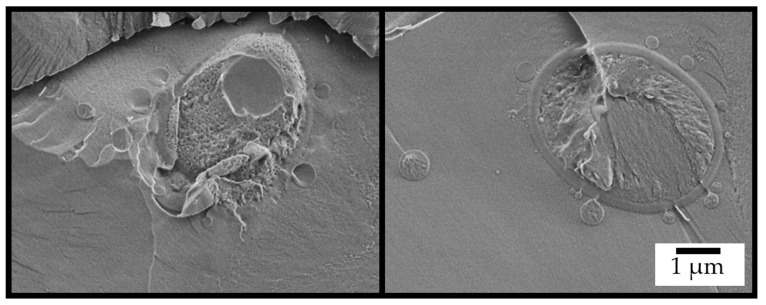
Cryo-SEM images of the fractured surface of the integrated system after 48 h of reaction.

**Figure 4 ijms-26-11902-f004:**
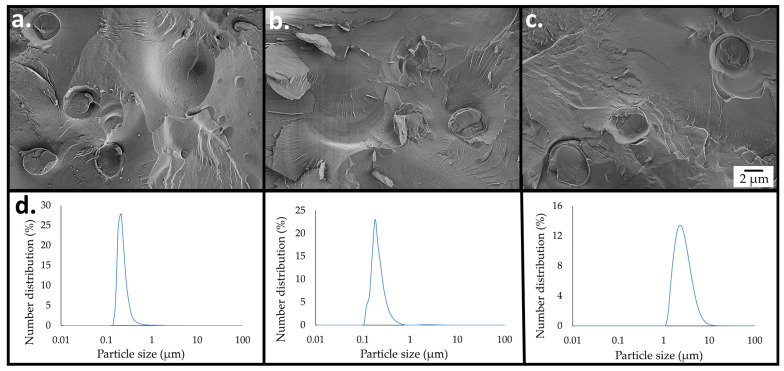
Progression of the cSFT process to generate emulsified cellulose-coated micro-particles integrated with enzymes and *S. cerevisiae* (1 wt. %), followed by cryo-SEM imaging of the cryo-fractured surface of the vitrified emulsion, after reaction durations of (**a**) 24 (**b**) 48, and (**c**) 72 h. (**d**) Particle size distributions (by number) evaluated by light scattering after reaction times, as shown in the images above each graph. The distributions evaluated at 24 and 48 h are dominated by micro-particles, whereas that after 72 h is due to the yeast cells.

**Figure 5 ijms-26-11902-f005:**
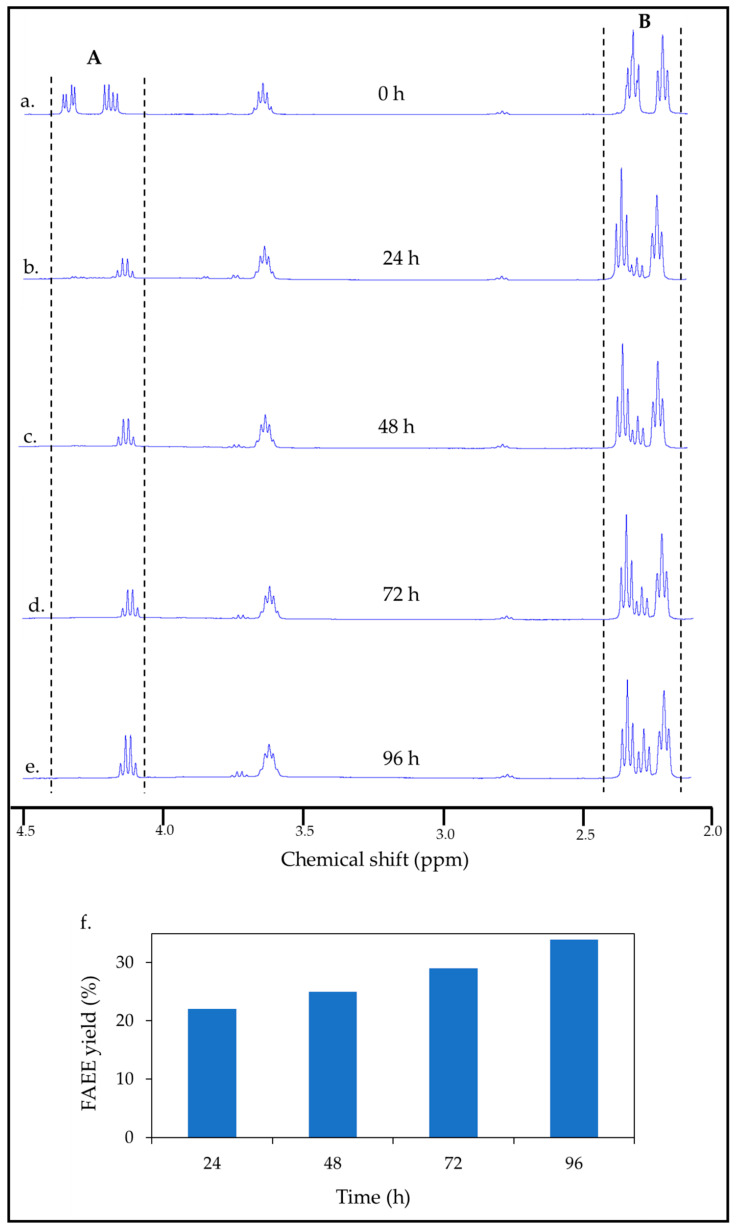
^1^H NMR spectra of (**a**) the cellulose-coated emulsion particle integrated with yeast. (**b**–**e**) The cSFT process over 96 h. Region A: peaks in the range of 4.28–4.32 ppm, refers to castor oil transitioning to FAEE; region B: peaks in the range of 2.24–2.38 ppm, refers to castor oil transitioning to FAEE and FFA. (**f**) FAEE yield of the cSFT process as a function of time.

**Figure 6 ijms-26-11902-f006:**
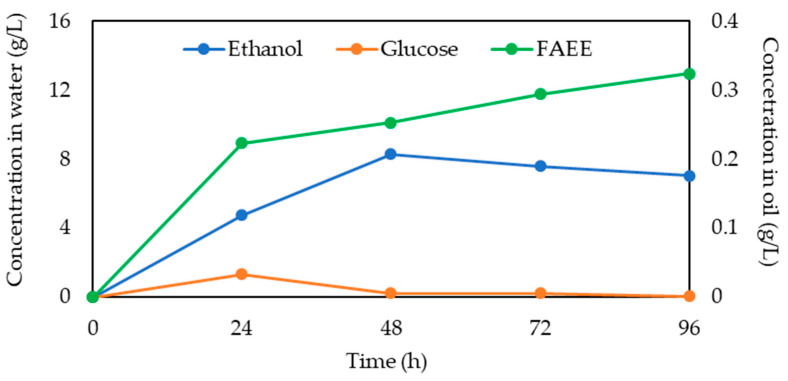
Glucose and ethanol concentration in water and FAEE concentration in oil over 96 h of the cSFT process.

## Data Availability

The raw data supporting the conclusions of this article will be made available by the authors upon request.

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
