# Peer review of "A Consolidated Saccharification, Fermentation, and Transesterification Process (cSFT) Converting Castor Oil to Biodiesel with Cellulose-Derived Ethanol"

_ijms, 2025, doi:10.3390/ijms262411902_

Round 1
Reviewer 1 Report
Comments and Suggestions for Authors
In this manuscript, titled "A consolidated saccharification, fermentation, and transesterification process (cSFT) for "one-pot" conversion of cellulose to biodiesel", the authors describe a one-pot system that integrates yeast with cellulose-coated emulsion microparticles, which are coupled with enzymes to act as a bioreactor for a series of biochemical reactions that transform cellulose into fatty-acid esters. While this approach is appealing from the perspective of green chemistry and sustainable development, there are several comments and questions about the effectiveness of the process and the methods used to conduct the study.
- The authors note that the FAEE yield is only 34%, which is a relatively low. This fact should be noted as a potential limitation for the practical application of the process.
- Furthermore, what is the isolated yield of FAEE for this system, and how easily will it be possible to separate the final products from the reaction mixture? Should the reaction be carried out until all starting and intermediate components have been completely converted? Also, add a section with economic calculations to determine the feasibility of the overall process. There are concerns about the cost-effectiveness of using enzymes.
- Could microorganisms at some point begin to use FAEE as a nutritional substrate and hydrolyze the end product back to ethanol?
- The authors note that other pre-treated biomass could also serve as a substrate for the reaction. In order to be thorough, the proposed cSFT system should be tested on other substrates, even without conducting a deep study, in order to demonstrate the feasibility of this reaction. This raises the question of whether other enzymes would be necessary, or if the proposed system would be capable of processing pre-treated biomass.
- It is worth considering the potential byproducts that could arise from such a process, as this multi-reaction "one-pot" system may be low selective. While model reactions and substrates may work well, real processes may encounter limitations.
- Saccharomyces cerevisiae and other microorganism names should be italicized. Is there more precise information about which strain was used? Is it regular baker's yeast?
- Line 20 and further: «cellulytic enzymes» should be probably replaced with «cellulolytic enzymes».
- Line 441: «g-1» should be replaced with «g-1».
- Line 444: Why were these specific cultivation conditions of 40 °C chosen? This temperature is almost the maximum for working with enzymes, and they will exhibit their greatest activity. However, it is also considered too high for cultivating microorganisms.
- Line 453: In this section, please provide a brief description of the method so that readers do not need to search for the source material.
- The time should be indicated in Figure 5.
Reviewer 2 Report
Comments and Suggestions for Authors
- The 34% FAEE yield after 96 hours is promising, yet the paper should discuss limiting factors like enzyme stability or mass transfer resistance. If possible, they can propose optimization strategies.
- Cryo-SEM provides valuable structural insights, but additional elemental mapping could clarify enzyme distribution on micro-particles and improve understanding of their spatial arrangement.
- The minimal cellulase dosage is a key strength, but the paper should elaborate on how this low dosage impacts reaction kinetics versus higher dosages.
- Using microcrystalline cellulose is a solid starting point. Testing pretreated lignocellulosic biomass like agricultural waste would demonstrate practical applicability and process robustness.
- The paper describes reaction progress over time, but modeling individual step kinetics would identify rate-limiting steps and guide optimization.
- The broad micro-particle size distribution may affect efficiency.
- Small-scale falcon tube experiments are foundational. Discussing scale-up challenges like emulsion stability in larger reactors and proposing scalable homogenization methods is necessary.
- The authors should enhance the literature review by adding more advanced studies on the current energy trend in the background, for example, https://doi.org/10.46690/compes.2024.01.01.
- Focusing on FAEE production is appropriate. Analyzing by-products like free fatty acids or glycerol would provide complete reaction selectivity data and inform minimization strategies.
Reviewer 3 Report
Comments and Suggestions for Authors
This paper describes a tranesterification of triglycerides of castor oil by biochemical way by ethanol obtained by fermentation of glucose derived from cellulose in the same biochemical reactor in one-pot procedure. The authors found an effective biochemical system for this mutual transformation. The process was thoroughly investigated by kinetic and spectral methods. This is a useful paper for biochemists and specialists in chemistry and technology of renewable plant materials.
However, the description of the data needs a major revision.
The comments are as follows.
- The title of the paper is not correct, since the authors did not convert directly cellulose to biodiesel. If they did that, they should get a Nobel Prize! In fact, the author used a mixture of cellulose and castor oil for this one-pot transesterefication of the latter. The title of the paper must be changed. The title must contain “castor oil”.
- In the Abstract, it is not correct to state that lingo-cellulosic biomass may be converted to fatty acid esters. In fact, lingo-cellulosic biomass, for instance, wood, does not contain a lot of fatty acid esters. The main sources of fatty acid esters, as triglycerides, are oils from oil plants, that are used for biodiesel production. This phrase must be corrected.
- The introductory section is too long. It must be reduced by at least half.
- The conclusion section must contain “castor oil” as well, see above the comment no. 1.
Reviewer 4 Report
Comments and Suggestions for Authors
This work presents a novel one-pot emulsion-based process transforming unmodified cellulose to biodiesel by hybridized of cellulose-coated micro-particles incorporating cellulytic enzymes and lipases with yeasts. Some results are obtained, while there are some problems.
- The background content is longer than the research content.
- The background of biomass should be strengthened by citing literatures, such as https://doi.org/10.1016/j.enchem.2024.100133.
- The quantitative data and corresponding discussions are lacked.
- What is the mechanism of one-pot emulsion-based process transforming unmodified cellulose to biodiesel by hybridized of cellulose-coated micro-particles incorporating cellulytic enzymes and lipases with yeasts?
Round 2
Reviewer 1 Report
Comments and Suggestions for Authors
The authors have responded to all questions and made the necessary revisions to the manuscript. After this revision, the paper may be published in its current form in the International Journal of Molecular Sciences.
Reviewer 3 Report
Comments and Suggestions for Authors
The paper may be accepted.
Reviewer 4 Report
Comments and Suggestions for Authors
Accepted.